# Mental health and psychosocial support strategies in highly contagious emerging disease outbreaks of substantial public concern: A systematic scoping review

**Angela M. Kunzler[1,2]\***, **Jutta Stoffers-Winterling[1,2]**, **Marlene Stoll[1,2]**, **Alexander L. Mancini[2]**, **Sophie Lehmann[1]**, **Manpreet Blessin[1]**, **Donya Gilan[1]**, **Isabella Helmreich[1]**, **Frank Hufert[3]**, **Klaus Lieb** [1,2]\*

**1** Leibniz Institute for Resilience Research (LIR), Mainz, Germany, **2** Department of Psychiatry and Psychotherapy, University Medical Center of the Johannes Gutenberg University, Mainz, Germany, **3** Institute of Microbiology and Virology, Brandenburg Medical School Theodor Fontane, Brandenburg, Germany

\* angela.kunzler@lir-mainz.de (AMK); klaus.lieb@lir-mainz.de (KL)

## Abstract

### Background

Acute disease outbreaks such as the COVID-19 pandemic cause a high burden of psychological distress in people worldwide. Interventions to enable people to better cope with such distress should be based on the best available evidence. We therefore performed a scoping review to systematically identify and summarize the available literature of interventions that target the distress of people in the face of highly contagious disease outbreaks.

### Methods

MEDLINE, Cochrane CENTRAL, Web of Science (January 2000 to May 7, 2020), and reference lists were systematically searched and screened by two independent reviewers. Quantitative and qualitative studies investigating the effects of psychological interventions before, during, and after outbreaks of highly contagious emerging infectious diseases, such as SARS, MERS, Ebola, or COVID-19 were included. Study effects were grouped (e.g. for healthcare professionals, community members, people at risk) and intervention contents at the individual and organizational level summarized. We assessed the level of evidence using a modified scheme from the Oxford Centre for Evidence-based Medicine and the Australian National Health and Medical Research Council.

### Results

Of 4030 records found, 19 studies were included (two RCTs). Most interventions were delivered during-exposure and face-to-face, focused on healthcare workers and crisis personnel, and combined psychoeducation with training of coping strategies. Based on two high-quality studies, beneficial effects were reported for resilience factors (e.g. positive cognitive appraisal) and professional attitudes of healthcare workers, with mixed findings for mental

**Data Availability Statement:** All relevant data are within the manuscript and Supporting Information files.

**Funding:** The authors (AMK, MS, KL) received funding from the German Federal Ministry for Education and Research (BMBF) as part of the Network for University Medicine (Grant number 01KX2021; CEOsys project). The funding body had no role in the design of the study and collection, analysis, and interpretation of data and in writing the manuscript.

**Competing interests:** AMK, MS, ALM, MB, and SL have no conflicts of interest. DG is in training as a board-certified cognitive-behavior (CB) therapist. JSW and IH are board-certified CB therapists. FH is board-certified in medical microbiology and virology and a specialist in tropical medicine (DTM&H) with a special interest in POCT-based rapid viral diagnostics. KL is a board-certified CB therapist with a special interest in schema therapy; he is also an Editor with the Cochrane Developmental, Psychosocial and Learning Problems Group. This does not alter our adherence to PLOS ONE policies on sharing data and materials.

health (e.g. depression). Across all studies, there was positive qualitative feedback from participants and facilitators. We identified seven ongoing studies mostly using online- and mobile-based deliveries.

## Conclusions

There is preliminary evidence for beneficial effects of interventions to enable people to better cope with the distress of highly contagious emerging disease outbreaks. Besides the need for more high-quality studies, the summarized evidence may inform decision makers to plan interventions during the current pandemic and to develop pandemic preparedness plans.

## Introduction

The COVID-19 pandemic, caused by severe acute respiratory syndrome coronavirus 2 (SARS-CoV-2) [1], has an enormous impact on healthcare and economic systems, but also on public mental health. On 30 January 2020, the WHO Director-General declared the novel coronavirus (2019-nCoV) outbreak a public health emergency of international concern [1].

Emerging new infections such as COVID-19 and pandemic outbreaks in general exert a significant psychological impact on healthcare staff and the community at large. Several recent reports have shown that especially healthcare workers are at risk of developing psychological distress and other mental health symptoms during the SARS-CoV-2 pandemic [2–6]. Possible causes may be overwhelming workload, lack of personal protection equipment, lack of intensive care unit beds and triage situations, lack of specific treatments, and feelings of not getting enough support from their superiors. A recent survey of healthcare workers in Wuhan and other regions in China demonstrated high levels of depression, anxiety, insomnia, and distress, with exceptionally high burden in healthcare workers who are directly in charge of patients with COVID-19 [4]. Surveys of the general population, mostly from China, where the pandemic started [1], have also shown that people show high amounts of anxiety, depressive, and stress symptoms [7–9]. Such distress may be caused by disease-related fears, social distancing and quarantine measures [10], and the potential long-term impact of pandemics (e.g. economic disruption) [11].

Fostering mental health and psychological adjustment (i.e. resilience) to the challenges of pandemics appears to be an important approach to prevent the development of mental disorders in both healthcare workers and the general population [12, 13]. The call for psychological interventions to cope with the immediate and long-term stress of pandemics had already been prompted after the SARS pandemic in 2003 and local Ebola epidemic outbreaks (e.g. 2014/2015 in West Africa) [14], but is becoming louder as the current pandemic is ongoing [15–17]. Epidemic and pandemic response plans also point to the need for measures of psychosocial support [18–20], and respective interventions have already been developed for SARS, influenza pandemics, and Ebola epidemics [21–23]. Since the WHO announced the COVID-19 outbreak a pandemic by March 2020, first interventions (e.g. online mental health services) to cope with the psychological impact of this disease were already initiated [24–26]. However, the currently available evidence of psychological interventions and potential benefits of training remain unclear.

In this scoping review, we intended to systematically examine relevant research in the field investigating strategies that have been implemented and evaluated to promote mental health and psychosocial support in people who are or were involved in the outbreak management of highly contagious infectious diseases, or are expected to be. In order to provide a reference point for policy makers and other deciders in clinical and public settings, we aimed to assess the

appropriateness, meaningfulness and feasibility of such interventions as well as their effects on mental health outcomes. Furthermore, we aimed to identify key concepts of such interventions which may serve as a starting point for the planning of crisis plans and future intervention studies.

## Materials and methods

### Search strategy

A protocol of this review was drafted according to the standards delineated in the Preferred Reporting Items for Systematic Reviews and Meta-analysis–scoping reviews extension (PRISMA-ScR) [27] and was registered at OSF registries in parallel (osf.io/5fevy). We searched three bibliographic databases from January 2000 to May 7, 2020: MEDLINE, Cochrane CENTRAL, and Web of Science. The search strategy combined three key sets of terms: functional ways of dealing with stress (e.g. resilience, psychological adjustment, coping), interventions (e.g. stress management), and disease outbreaks (e.g. SARS, pandemic). Various terms for each of the concepts were entered, as appropriate for each database (e.g. as MeSH terms, keywords, text words). The full search string for all databases is outlined in the S1 Appendix. The search was limited to studies of humans. There were no restrictions concerning language, publication status, date, or format. The reference lists of eligible studies and of relevant sources (e.g. reviews) were hand searched to identify any additional relevant studies.

### Eligibility criteria

We included individuals in anticipation of, during, or in the aftermath of an outbreak of a highly contagious emerging disease, irrespective of age and health. Eligible interventions were any psychological interventions to foster the adjustment of individuals (possibly) exposed to pandemics or hazardous infectious disease outbreaks and/or to promote their mental health and psychosocial support. We included quantitative and qualitative studies (for detailed eligibility criteria see S1 Table).

### Study selection and data analysis

After de-duplication, the screening and study selection process were performed in duplicate by two independent reviewers (AMK and MS, ALM, MB, SW, respectively). Titles and abstracts were screened to exclude evidently irrelevant records, then full-text records of relevant studies were inspected to determine eligible studies. Any disagreements were resolved by discussion or by consulting a third reviewer (KL). If necessary, study authors were contacted. Discrepancies were resolved at each stage and consensus was achieved with excellent inter-rater reliability (κ = 0.84). Two independent reviewers (AMK, KL) extracted relevant information for each included study using a customized spreadsheet. Intervention contents at the individual and organizational level across eligible studies were also extracted by two reviewers (SW, KL) working independently. Discrepancies in data extraction were resolved through discussion or by a third reviewer (JSW, DG, IH).

 The quality of the evidence of completed and published studies was assessed by two independent reviewers (AK, KL) using a rating scheme that was modified from the Oxford Centre for Evidence-based Medicine (EBM) for ratings of individual studies [28] and the Australian National Health and Medical Research Council (NHMRC) evidence hierarchy [29]. Given the status of this work as a scoping review, which do not require a formal assessment of the methodological quality of included studies [30], we used a rating scheme with four levels in order to assess the level of evidence (level 1: randomized controlled trial; level 2: controlled trial without randomization, case-control study or controlled time series design; level 3: single-group study

with either posttest or pretest/posttest assessment; level 4: case study/report with either posttest or pretest/posttest assessment).

We did not quantitatively integrate research findings because of the variety of included interventions, settings, and outcomes. Instead, we aimed at giving a comprehensive overview of intervention concepts based on the available quantitative and qualitative evidence. To do so, we first grouped the publications by target populations (healthcare professionals and crisis personnel; children and adolescents, community members, patients with highly contagious infectious disease, and people at elevated risk due to somatic conditions), and the time point of implementation (pre-exposure, during-exposure, after-exposure), respectively. We summarized the quantitative and qualitative findings for these interventions, if available. Second, we examined the intervention contents of eligible studies and additional observations described by the publications (e.g. feedback from participants) in order to give an overview of key concepts used in the respective interventions relating to the individual and organizational level. Overall, this systematic scoping review adheres to the PRISMA-ScR statement [27] (see S2 Table).

## Results

The literature search yielded 4030 records (Fig 1).

After de-duplication, 3872 records were retained for title/abstract screening. Of these records, 149 were assessed at full-text level. The full-text screening resulted in 19 completed

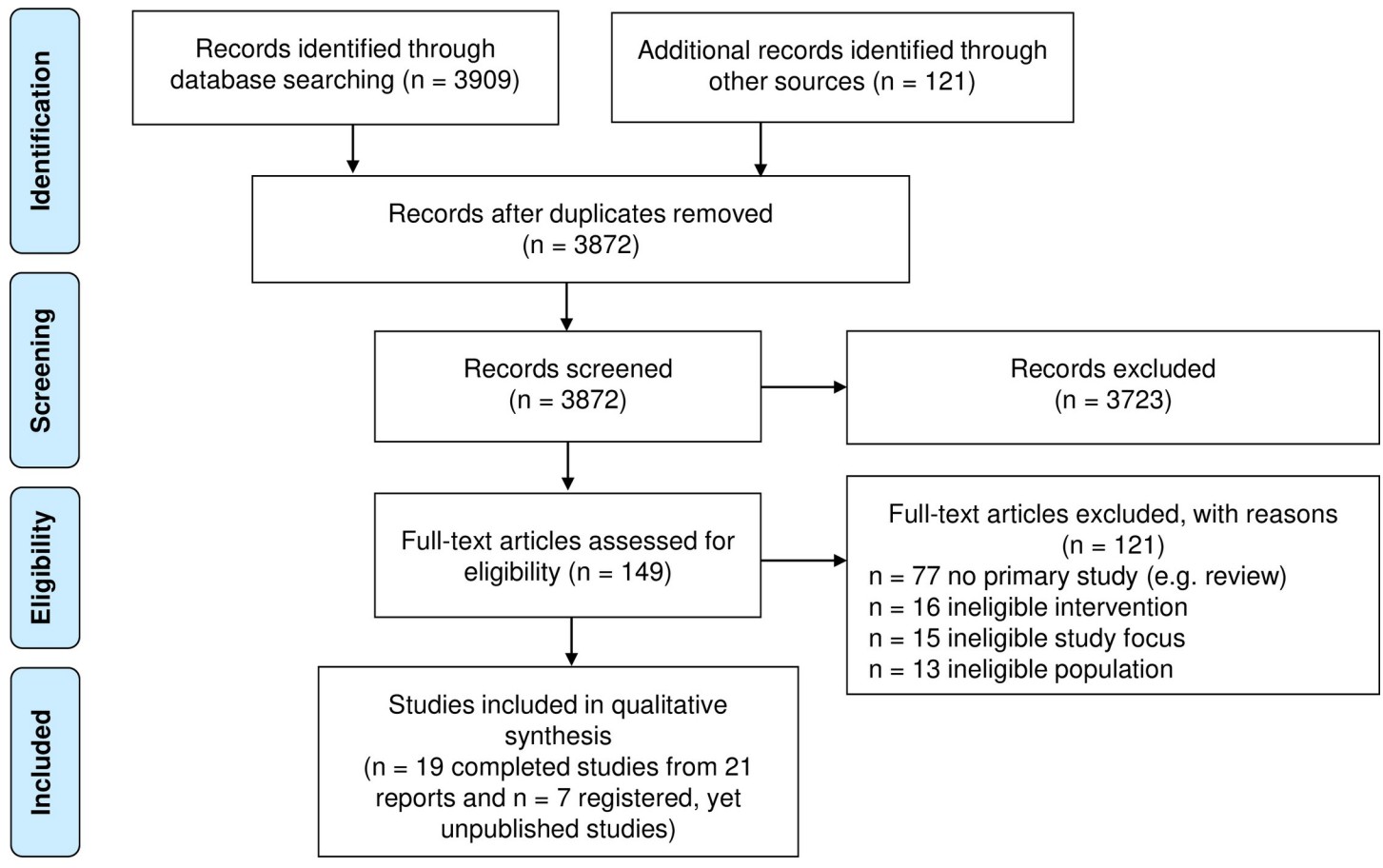

**Fig 1. PRISMA flow diagram.** Searched Jan 2000 to May 7, 2020; last search date: May 7, 2020.

studies (from 21 reports) that were included in the review. The study characteristics of these studies are given in Table 1. We also identified seven eligible registered studies (see Currently registered, yet unpublished intervention studies).

Among the completed and published studies, quantitative data were measured in nine studies. We found five qualitative studies, four studies used a mixed-methods design, one case report provided no data [26].We identified three trials including parallel controls (one randomized study [31], one RCT [22], and one cluster-RCT [32]), 11 single-group studies [21, 33–41, 47, 48] including one interview study [47], four case reports/studies [23, 26, 42, 43], and one two-group study (delayed treatment model) [44, 45]. Most of the 19 eligible studies were performed in adults (total of 4775 participants), with only two studies investigating a sample of children or adolescents (total of 600 participants) [40, 44–46]. Two studies did not examine the participants of training, but performed qualitative interviews with facilitators [47, 48]. Across the 19 studies, sample sizes ranged between 12 and 1250 participants. The quality of the evidence ranged between level 1 and level 4, with two level-1 studies that were RCTs [22, 32], two (level 2) controlled studies without randomization [31, 44, 45], 11 single-group studies of level 3 [21, 33–41, 47, 48], and four case studies/reports of level 4 [23, 26, 42, 43].

## Studies in healthcare professionals and crisis personnel

Healthcare professionals and crisis personnel (e.g. local hospital or treatment center staff, deployed individuals providing help on-site)–that is, individuals directly exposed to acute disease outbreaks–were most frequently addressed by interventions, with 12 studies performed in this target group.

In four studies that aimed to prepare healthcare staff and crisis personnel for an *anticipated* pandemic, participants were reported to feel better able to cope compared to before the training [21]. The randomized study of Maunder and colleagues [31], who compared different doses of resilience training, suggested evidence of increases in resilience factors (e.g. pandemic self-efficacy), the confidence in training, and interpersonal problems. For two pre-deployment trainings focusing on resilience (e.g. coping, resilience factors like problem-solving), combined with virtual reality, Klomp and colleagues [34, 35] reported improvements in resilience and self-efficacy.

Five interventions were offered *during* a pandemic, with one crisis intervention also referring to quarantined medical staff during the current COVID-19 pandemic [26]. Regarding the effectiveness of Psychological First Aid (PFA) to equip volunteers to provide psychosocial support, the results of another study were mixed [47]. In a safety training also containing psychological elements (e.g. stress management), participants reported to benefit most from hands-on aspects [37]. In a qualitative study, emergency hotline workers reported to find a support group helpful for, for example, stress management and self-efficacy [33]. A pre-incident stress inoculation training (e.g. personal resilience plan, monitoring of own stressors) resulted in 90% participants under the clinical cut-off for a posttraumatic stress disorder in a self-triage encounter system [23].

The only cluster-RCT comparing PFA to wait-list control for healthcare staff *after* the 2014 Ebola outbreak found mixed results, with small effects on professional attitude versus no effects on, for example, professional quality of life [32]. Waterman and colleagues [39] examined the only stepped care approach in this review. For a three-phase intervention (e.g. CBT group intervention), developed to be implemented during or after an Ebola outbreak, the authors reported evidence of positive training effects for several mental health outcomes (e.g. depression, posttraumatic stress symptoms). In a qualitative study [48] with facilitators of the CBT group, the emerging social networks and the novelty of CBT were highlighted as key

**Table 1. Characteristics of included studies.**

| Study (year); country | Pandemic; time point | Population | Number of included subjects (mean age (SD), % females)[a] | Study design (quality of the evidence level)[a] | Intervention name; setting; delivery; intensity; theoretical approach; intervention providers; control group (if available) | Intervention content | Evaluation | Psychological outcomes (scale) and effects |
|---|---|---|---|---|---|---|---|---|
| **Interventions for healthcare professionals and crisis personnel** | | | | | | | | |
| Aiello et al. (2011), Canada | H1N1 influenza; before | Hospital staff (22 departments) | N = 1250; NA; NA | Single-group study; mixed-methods (level 3) | *Pandemic resilience training session*: group (n = 5–50); face-to-face; one 1-hour session; Folkman and Greer's model of coping; two members of a Psychosocial Pandemic Committee | Information on stress, coping and resilience strategies, organizational issues | Quantitative post-session evaluation using feedback surveys (8 items; e.g. compared to situation prior to the session); additional qualitative feedback | 76% felt better able to cope after training as compared to 35% prior to training; 10 key qualitative themes from participants' comments (e.g. family-work balance) |
| Schreiber et al. (2019), West Africa | Ebola outbreak in 2014/2015; during | US healthcare workers deployed to Africa | N = 45; 25–60 years old; NA | Case report; mixed-methods (level 4) | *Anticipate, Plan, and Deter (APD) Responder Risk and Resilience model*: NA; NA; NA; NA; instructors with completed APD "train the trainer" education | Anticipate: pre-event stress inoculation training; Plan: development of "personal resilience plan"; Deter: use personal resilience plan; monitoring of own stress exposure | Healthcare workers encouraged to complete daily self-triage of their risk factors for traumatic stress during past 24 hours using PsySTART-R system; retrospective qualitative analysis of training | Aggregated PsySTART-R triage encounters: 90% were under the cut-off score for a clinical diagnosis of PTSD |
| Zhang et al. (2020), China | COVID-19 outbreak in 2019; during | Quarantined population (patients/family) and medical staff | NA; NA; NA | Case report with description of implementation in local hospital (level 4) | *Psychological crisis intervention*: individual; online consulting (WeChat) and telephone hotline; online and hotline service all day long; NA; self-administered and/or consultation with physician, psychological consultant, psychiatrist; *APD Responder Risk and Resilience model* | Self-management: popular science reading, health education (e.g. deal with information explosion); mental health self-evaluation; self-aid skills training (breathing relaxation, deal with virus-related anxiety, mindfulness-based stress reduction) | NA | NA |
| Maunder et al. (2010), Canada | Influenza; before | Hospital workers (all employees and professional staff) | N = 158; NA; NA for participants randomized (228 [86%] of those who consented to participate) | Randomized trial (level 2) | *Resilience training "Pandemic Influenza Stress Vaccine"*: individual; computer-assisted; three courses of different length (7, 12, 17 sessions); NA; self-administered with interactive exercises between participant and researcher | Pandemic-related knowledge provision, information on stress, relaxation techniques, coping strategies, training of self-efficacy, management of interpersonal problems (e.g. family-work balance) | Questionnaire on confidence in training and support (modified for influenza); Pandemic Self-Efficacy Scale; Inventory of Interpersonal Problems (IIP-32); Ways of Coping inventory | Significant improvements in confidence in support and training, pandemic self-efficacy and interpersonal problems, the last especially in medium and long courses; not significant in coping |
| Sijbrandij et al. (2020), Sierra Leone | Ebola outbreak in 2014/2015; after | Healthcare workers at Peripheral Health Units (PHU) | N = 129 PHUs (N = 408 healthcare workers, IG: n = 202, CG: n = 206); 39 (9.13); 341 (84%) females[b] | Cluster-randomized controlled trial (level 1) | *Psychological First Aid (PFA)*: group (n = 19); face-to-face; 1-day training; based on PFA Training of Trainers manual adapted by WHO Mental Health focal person for Sierra Leone; mental health nurses; CG: wait-list control | Information on stress, mental health/ disorders, psychosocial support; information on PFA; self-care; provision of PFA (e.g. practicing PFA with role-play) | Self-report instruments, professional attitude, confidence, and professional quality of life at baseline, posttest (shortly after PFA training) and 3-month follow-up | Follow-up: small effects for knowledge, understanding of applying PFA, professional attitude; no effects for confidence and professional quality of life at any assessment |
| Kahn et al. (2016), Sierra Leone | Ebola outbreak in 2014/2015; during | Emergency hotline workers | N = 44 of 150; NA; NA | Single-group study; qualitative interviews (level 3) | *Support group*: group; face-to-face; weekly group sessions from December to May 2015; key elements: debriefing, peer support, self-care, and psychoeducation; psychosocial specialists affiliated with IsraAID | Specific focus each week: e.g. trauma, team building and coping strategies (e.g. relaxation, self-advocacy) | Interviews using standardized open-ended approach with 44 hotline workers (feedback about and ways of benefitting from support group) | Helpful to manage abusive calls (n = 14); for stress management (n = 10); for greater self-efficacy and increased capacity for empathy (n = 8); to develop confidence to perform job responsibilities (n = 5) |
| Klomp et al. (2020), USA | Ebola outbreak in 2014/2015; before (in pre-deployment training) | Individuals deployed by the CDC | N = 100; NA; NA | Single-group study (level 3) | *Deployment Safety Resiliency Team (DSRT)*: group; face-to-face and VR; 3-day training; PFA; five experienced trainers from the CDC's Resilience Assessment and Maintenance Program | PFA principles (peer support, coping skills, stress management, triage, proper referral processes); Disaster Site Safety (e.g. personal protective equipment); Virtual Reality Environment (VRE) experience | DSRT Course Content Survey (e.g. mastery of constructs related to PFA, DSRT principles, resilience, support, referral recommendations); 10-item General Self-Efficacy Scale at baseline and posttest | Significant improvement in DSRT Course Content Survey for effectiveness of training and in Self-Efficacy Scale between baseline and posttest |
| Klomp et al. (2011), USA | Disease outbreaks in general, before | Individuals before deployment by the United States Centers for Disease Control and Prevention (CDC) | N = 22; NA; NA | Single-group study; mixed-methods (level 3) | Virtual Classroom Immersion Training: group, face-to-face and VR; 3-day course with 2-day resiliency component; PFA, stress management, coping strategies; NA | Resiliency component: role clarification (PFA, resilience factors, e.g. self-efficacy); core actions like stabilization, social support, information on coping (e.g. resiliency, coping, relaxation, stress management, problem-solving) | Preliminary program evaluation concerning knowledge, skill, and attitudes; qualitative feedback from participants | Posttest average score for resiliency component of DSRT 83% compared to 77% before; positive qualitative feedback: better response in actual situations, felt better prepared for challenges of deployment |

(Continued)

**Table 1.** (Continued)

| Study (year), country | Pandemic; time point | Population | Number of included subjects (mean age (SD), % females) | Study design (quality of the evidence level)[a] | Intervention name; setting; delivery; intensity; theoretical approach; intervention providers; control group (if available) | Intervention content | Evaluation | Psychological outcomes (scale) and effects |
|---|---|---|---|---|---|---|---|---|
| Narra et al. (2017), USA | Ebola outbreak in 2014/2015; during | US healthcare workers deployed to work in West Africa ETUs | N = 570; NA; NA | Single-group study; qualitative (level 3) | *EVD Safety Training Course:* group; face-to-face (classroom instruction and hands-on training in mock ETU); 3-day course; drawn from materials of Médecins sans Frontières, WHO, CDC; NA; multidisciplinary team (e.g. physicians) | Psychological elements of course: psychological preparation, stress management (mental health resilience, occupational health, community health promotion) | Qualitative feedback from course graduates and returning responders | Hands-on aspects of course most important |
| Waterman et al. (2018); Waterman et al. (2019), Sierra Leone | Ebola outbreak in 2014/2015; during/after | Ebola Treatment Centre (ETC) staff | N = 75 of 3273 who entered all three phases; 29.46 (7.40); 641 (19.6%) females (for 1135 (34.7%) NA) | Single-group study (level 3) | *3-phase intervention:* • *PHASE 1: Well-being Screening Workshop:* group (max. n = 50); face-to-face; 2-hour workshop; based on PFA • *PHASE 2: Psychoeducational workshops:* group; face-to-face; 2-hour workshop; NA • *PHASE 3: Cognitive-behavioral therapy (CBT)-based group intervention:* group (n = 14); face-to-face; six weekly sessions; CBT; for all phases: peer-delivered by ex-Ebola ETC staff previously trained in CBT intervention by UK clinician | • PHASE 1: Discussion of challenges of work and coping styles • PHASE 2: focus on common mental health difficulties psychoeducation; coping strategies • PHASE 3: low-intensity CBT program with behavioral activation, minimizing avoidance, problem-solving, coping with anxiety | • PHASE 1: Well-being screening • PHASE 2: single item Well-being screening; for different workshops: Post-Traumatic Stress Checklist (PCL-C); Perceived Stress Scale (PSS); Insomnia Severity Index (ISI); Generalized Anxiety Disorder 7 (GAD7); Patient Health Questionnaire-9 (PHQ-9); Relationship Questionnaire; behavior problems, Dimensions of Anger Reaction (DAR-5), Alcohol Use Disorders Identification Test-C • PHASE 3: all measures at start and two weeks after completion | • Across all phases: significant improvements in Well-being Screening Measure for stress, depression, anxiety, behavior, relationships • PHASE 2: significant improvements for stress, anxiety, depression, behavioral problems, alcohol usage • PHASE 3: significant improvement in Well-being Screening Measure and clinical measures for PTSD, stress, sleep, anxiety, depression, anger, relationship difficulties |
| Cole et al. (2020), Sierra Leone | Ebola outbreak in 2014/2015; after | Former ETC staff with symptoms of anxiety and depression | N = 253; 30 (7.04); 58 (22.9%) females | Single-group study (level 3) | *Group CBT for depression and anxiety program:* group; face-to-face supplemented by booklet; six weekly 3-hour sessions; CBT; two Sierra Leonean former ETC staff with training in CBT group delivery | Behavioral activation and reducing avoidance behaviors; identifying values and generating related goals; ways to deal with unhelpful cognitions and thinking patterns; problem-solving skills; strategies to manage anxiety | Pre-post assessment with GAD7, PHQ-9, Work and Social Adjustment Scale (WSAS) for functional impairment, feedback questionnaire for participants' experiences with intervention one-week pre-intervention and within two weeks post-intervention | Significant improvements for three outcomes; no effect of ETC role risk on improvement over time; intervention helped to personal goals or recovery 'a great deal' (60.8%) |
| Horn et al. (2019), Sierra Leone and Liberia | Ebola outbreak in 2014/2015; during | Non-specialist volunteers | N = 36 providers previously trained in PFA; NA; 13 (36%) females | Qualitative interview study (level 3) | *Psychological First Aid (PFA):* group; face-to-face; 1-day training; evidence-informed general principles for successful interventions (promote sense of safety, calming, sense of self and collective efficacy, connectedness, hope); NA | Staying calm (e.g. focus on breathing), identify/prioritize needs, training of self-efficacy and positive coping strategies, connecting with loved ones and social supports, focus on strengths and coping ability, spirituality | Interviews with providers (how was PFA approach understood and used), but no evaluation of training effects | Some hints that PFA is effective in volunteers to provide psychosocial support: calming element particularly important, promotion of sense of self and collective efficacy and of connectedness particularly challenging |
| **Interventions for children and adolescents** | | | | | | | | |
| Yau et al. (2004) described in Chan et al. (2006), Hong Kong | SARS outbreak in 2003; after | Adolescents at junior high school (grade eight) | N = 244; NA; NA | Single-group study (level 3) | *Strength-Focused and Meaning-Oriented Approach for Resilience and Transformation (SMART):* group; face-to-face; 1-day psychoeducational program; holistic intervention, body-mind spirit model; NA | Pandemic- and SARS related knowledge (e.g. symptoms); Eastern spiritual teaching; psychoeducation on cognitive reappraisal, dynamic coping (e.g. with fear), emotional well-being, meaning making; yoga and meditation | NA | Significant increase of sense of social commitment, mastery of life, learning and growth; significant decrease of sense of social disintegration and loss of security |
| Decosimo et al. (2019); Decosimo et al. (2017), Liberia | Ebola outbreak in 2014/2015; during | Children (Ebola survivors, from Ebola-infected homes, living in Ebola-affected community) | N = 356 from 870 (IG1: 533; IG2: 337); 9.97 (3.67); 220 (61.8%) females | Two-group study after two pilot phases with delayed treatment model (IG2 starting two months after IG1); mixed-methods (level 2) | *Community-based psychosocial expressive arts program ("Playing to Live"):* group; face-to-face; two intensities (5 months, 3 months; at least 2–3 activities per week); theory from the fields of expressive arts therapy; trained para professionals (Ebola survivors) | Building social support; teaching child specific trauma coping skills, building sage space for children to express themselves | Pre-post psychological stress symptoms (PSS) by child interviews in 16 participating communities; facilitator's perspectives and child participant quotes (127 randomly selected reports) | Pre-post decrease of reported stress symptoms and total symptoms in both groups; no difference longer vs shorter program; facilitator perspectives: 22.8% happy children, 4.7% sad children; 45.7% children active/ working, 5.5% positive reactions; child quotes: 18.1% reflect positive emotions, 14.2% with desires (e.g. hope) |

*(Continued)*

**Table 1.** (Continued)

| Study (year); country | Pandemic; time point | Population | Number of included subjects (mean age (SD), % females) | Study design (quality of the evidence level)[a] | Intervention name; setting; delivery; intensity; theoretical approach; intervention providers; control group (if available) | Intervention content | Evaluation | Psychological outcomes (scale) and effects |
|---|---|---|---|---|---|---|---|---|
| **Interventions for community members** | | | | | | | | |
| Morelli et al. (2019), Liberia | Ebola outbreak in 2014/2015; after | Community members of town | N = 60; NA (mixed sexes and ages) | Single-group study; qualitative (level 3) | *Social Reconnections Groups (community-based mental health and psychosocial group)*: group (four small groups with n = 14–17); face-to-face; 10 weekly half-day sessions; sociotherapy; staff from International Medical Corps (IMC) | Reflection on personal experiences with the outbreak, social support building, dealing with grief reactions, anger and conflict management, identification of values and strengths, coping strategies, dealing with past and future | Qualitative feedback from participants; observations made by facilitators during group sessions | Positive and critical comments received from participants (e.g. helped to overcome grief and go back to normal activities) |
| Pan et al. (2005), Taiwan | SARS outbreak 2003, during | Home-quarantined college students | N = 12; age range: 20–24 years; six (50%) females | Single-group study; qualitative (level 3) | *Support Group*: group; face-to-face; five sessions (500 minutes); therapeutic and curative factors for group counseling and therapy; licensed counseling psychologist, counselor in rehabilitation and senior counselor | Structured human relation activities; SARS-related knowledge; reflection on personal life stressors (e.g. timeline) and vulnerabilities; training of self-efficacy | Observations made from investigators during implementation; no evaluation of training effects | Sense of connection within group most significant experience; group cohesiveness and self-disclosure difficult to establish, especially for people suffering from trauma; positive group climate for exploration |
| Yoon et al. (2016), South Korea | MERS outbreak 2015, during | People in quarantine and families of MERS patients | N = 1221 individuals in quarantine; NA, NA | Case report (level 4) | *Mental health service system for MERS victims*: 1) mental health service for people placed in quarantine; 2) service for recovered patients and family members of deceased patients; NA; face to face; NA; public mental health and community mental health centers | NA | Service utilization rate for mental health service | Of 1221 people with emotional disturbances, 871 (71.3%) received one consultation, 350 (28.7%) required continuing services; 124 (2% of quarantined individuals) received continued services with average of 4.7 consultations |
| **Interventions for patients with highly contagious infectious disease** | | | | | | | | |
| Weissbecker et al. (2018), Sierra Leone and Liberia | Ebola outbreak in 2014/2015; during | Ebola patients admitted to Ebola treatment units (ETU) | N = 470 (303 with available data); age range: 0–≥55; 189 (62.4%) females | Case study (level 4) | *Psychosocial support (PSS)*: combined (one-to-one PSS most frequently); face-to-face; NA; PFA adapted to Ebola context and relevant principles of IASC Guidelines on Mental Health and Psychosocial Support in Emergencies; paraprofessional PSS officers (e.g. ETU staff) with previous training and Ebola survivors | Psychosocial support measures in ETU environment and procedures to minimize distress and enhance well-being, dignity and social connections in patients and relatives; health and hygiene education; nonpharmacological symptom management along with stress management (e.g. relaxation exercises) | Assessment of mental health problems during reports of psychosocial support activities | Decline of low mood and appetite problems during stay in ETU, no effect on anxiety and worry |
| **Interventions for people at elevated risk due to somatic conditions** | | | | | | | | |
| Ng et al. (2006), Hong Kong | SARS outbreak in 2003; after | People with chronic diseases | N = 51 (IG: n = 25, CG: n = 26); 55.28 (12.94); 38 (74.5%) females | Randomized controlled trial (level 1) | *Strength-Focused and Meaning-Oriented Approach for Resilience and Transformation (SMART)*: group; face-to-face; 1-day psychoeducational program; adaptation of critical incident stress debriefing protocol, holistic intervention with concepts of Daoism and Buddhism; NA; CG: no intervention | Pandemic- and SARS related knowledge (e.g. symptoms), psychoeducation on cognitive reappraisal, coping of fear, emotional well-being, meaning making | Self-developed scale to measure cognitive appraisal of SARS outbreak; subscales of Brief Symptom Inventory (BSI) for anxiety, depression, somatization, hostility at baseline, posttest and 1-month follow-up | Posttest: significant within-group decreases for depression, personal-negative and social-negative appraisal in IG; 1-month follow-up: significant between-group effect for depression, personal-positive and social-negative appraisal |

APD, Anticipate, Plan, and Deter; CBT, Cognitive Behavioral Therapy; CDC, Centers for Disease Control; CG, Control Group; DSRT, Deployment Safety Resiliency Team; ETU, Ebola treatment units; EVD, Ebola Virus Disease; IASC, International Accounting Standards Committee; IG, Intervention Group; MERS, Middle East Respiratory Syndrome; NA, not applicable/assessed; n, sample size; PFA, Psychological First Aid; PHU, Peripheral Health Units; PSS, Psychosocial Support; PTSD, Posttraumatic Stress Disorder; PsySTART-R system, Psychological Simple Triage and Rapid Treatment–Responder self-triage tool; SARS, Severe Acute Respiratory Syndrome; US, United States; VR, Virtual Reality.

[a] Evidence level according to the modified rating scheme from the Oxford Centre for Evidence-based Medicine and the Australian National Health and Medical Research Council.

[b] Data on gender only available for 407 participants.

enablers, while different cultural concepts of mental health problems presented key barriers. Finally, when evaluating the same CBT group in former Ebola Treatment Center (ETC) staff, positive effects on anxiety, depression and functional impairment were reported, with the effects being independent of the exposure to risk of infection during the ETC work [41].

## Studies in community members and other groups

Four studies focused on community members (e.g. people in quarantine). Based on qualitative data, a Social Reconnections Group helped participants to overcome grief and return to normal life [36]. From two studies in quarantined individuals [38, 42], Pan and colleagues made mixed observations from a support group during the SARS outbreak (e.g. sense of connection is important, self-disclosure difficult). Yoon and colleagues [42] described the implementation of a mental health service system for people in quarantine and families of patients, with 2% of quarantined individuals receiving continued services. For one intervention implemented during the current coronavirus outbreak [26], no data are available yet.

For both interventions focusing on children and adolescents during and after SARS and Ebola outbreaks (e.g. strength-focused and meaning-oriented resilience training) [40, 44–46], positive quantitative findings were reported (e.g. reduction of psychological stress, increase of sense of social commitment and resilience factors), in part supported by positive qualitative feedback.

One study focused on patients with a highly contagious infectious disease by investigating patients during the 2014 Ebola outbreak [43]. For psychosocial support measures in Ebola treatment units, including, for example, measures to minimize distress and enhance well-being as well as stress management, the authors found mixed effects with a decline of low mood compared to no effects on other measures (e.g. anxiety).

People at elevated health risk due to somatic conditions–a frequently named risk group during the current COVID-19 pandemic–were subject to one study. Ng and colleagues [22] compared a resilience training (same intervention as Yau and colleagues [40]) to no-intervention control. While there was evidence in favor of training for depression and cognitive appraisal of the SARS outbreak, the study found no effects on other mental health outcomes (e.g. anxiety).

## Contents of intervention studies

The 19 studies qualitatively described important contents that should be addressed in interventions to cope with the psychological challenges of a pandemic. These were immediately focused in the trainings or resulted from the observations made during the implementation. The recommended targets on the individual and organizational level are summarized in Tables 2 and 3, respectively.

## Currently registered, yet unpublished intervention studies

In addition to the above-mentioned completed studies, seven trials to cope with the psychological impact of the COVID-19 pandemic have been registered recently, all of them RCTs [49–55] (one with cross-over design) [49]. In contrast to the completed studies, six of these trials used a computer-, online- or mobile-based format to deliver the intervention [49–53, 55]. Various populations are targeted, including the general population [49, 50], different patient populations (e.g. obstetrics/gynecology patients) [51–53] or certain risk groups for mental and physical health problems (e.g. older individuals) [54, 55]. Besides interventions providing general health education (e.g. on mental health) [54], the training programs included mindfulness practices (e.g. with yoga) [50, 51, 55], CBT techniques (e.g. to deal with dysfunctional worry

**Table 2. Contents of interventions at the individual level.**

| Intervention content | Examples | Study population | | | |
|---|---|---|---|---|---|
| | | 1[a] | 2[b] | 3[c] | 4[d] |
| **General health care information** | Health and hygiene information, help system; clinical/site structure | Horn et al. (2019); Kahn et al. (2016); Klomp et al. (2011); Klomp et al. (2020); Maunder et al. (2010); Narra et al. (2017); Zhang et al. (2020) | Yau et al. (2004) | Morelli et al. (2019) | Weissbecker et al. (2018) |
| **Psychoeducation about stressors and mental health problems** | Common concerns; impact on mental health; sources and signs of stress | Aiello et al. (2011); Klomp et al. (2011); Maunder et al. (2010); Schreiber et al. (2019); Sijbrandij et al. (2020); Waterman et al. (2018); Zhang et al. (2020) | Decosimo et al. (2017); Decosimo et al. (2019) | Morelli et al. (2019); Pan et al. (2005) | Ng et al. (2006) |
| **Training/provision of coping skills** | Coping skills training; problem-solving, CBT strategies; stress management techniques; anger and conflict management | Aiello et al. (2011); Horn et al. (2019); Kahn et al. (2016); Klomp et al. (2011); Klomp et al. (2020); Maunder et al. (2010); Narra et al. (2017); Sijbrandij et al. (2020); Waterman et al. (2018); Waterman et al. (2019); Zhang et al. (2020) | Decosimo et al. (2017); Decosimo et al. (2019) | Morelli et al. (2019); Pan et al. (2005) | Weissbecker et al. (2018) |
| **Training of specific resilience factors** | (Psycho-)social support (e.g. involving family, peer support); religiosity/ spirituality; promote hope; meaning/ purpose in life (e.g. training of strengths and values); self-efficacy; optimism; self-esteem | Aiello et al. (2011); Horn et al. (2019); Klomp et al. (2011); Klomp et al. (2020); Maunder et al. (2010); Schreiber et al. (2019); Sijbrandij et al. (2020); Zhang et al. (2020) | Decosimo et al. (2017); Decosimo et al. (2019); Yau et al. (2004) | Morelli et al. (2019); Pan et al. (2005) | Ng et al. (2006); Weissbecker et al. (2018) |
| **Prevention of trauma, depression, and anxiety** | Behavioral activation; minimizing avoidance; CBT based training; arts therapy | Klomp et al. (2011); Maunder et al. (2010); Sijbrandij et al. (2020); Waterman et al. (2018) | Decosimo et al. (2017); Decosimo et al. (2019) | | |
| **Fostering sense of control** | Confidence in support system and well-prepared hospitals; command courses | Klomp et al. (2020); Maunder et al. (2010) | | | |
| **Relaxation, mindfulness and meditation techniques** | Relaxation breathing; muscle relaxation; Tai Chi; acupressure | Horn et al. (2019); Maunder et al. (2010); Kahn et al. (2016); Klomp et al. (2011); Klomp et al. (2020); Zhang et al. (2020) | Yau et al. (2004) | Morelli et al. (2019) | Ng et al. (2006); Weissbecker et al. (2018) |
| **Psychological First Aid (PFA)** | 5 core principles (safety; calming; connectedness; self-efficacy; hope/ optimism) and eight core actions (contact and engagement; safety and comfort; stabilization; information gathering; practical assistance; connection with social supports; information on coping; linkage with collaborative services) | Horn et al. (2019); Klomp et al. (2011); Klomp et al. (2020); Sijbrandij et al. (2020); Waterman et al. (2018); Zhang et al. (2020) | | Morelli et al. (2019) | Weissbecker et al. (2018) |
| **Dealing with death** | Dealing with grief reactions | | | Morelli et al. (2019) | Weissbecker et al. (2018) |

[a]1, healthcare professionals and crisis personnel

[b]2, children and adolescents

[c]3, community members

[d]4, patients with highly contagious infectious disease and people at elevated risk due to somatic conditions.

related to COVID-19) [49] and Acceptance and Commitment Therapy [53], or a combination of different techniques [52].

## Discussion

With this systematic scoping review, we identified interventions to provide psychosocial and mental health support in different target groups in the face of highly contagious infectious disease outbreaks. Evidence of positive effects on psychological outcomes was mostly reported by

**Table 3. Contents of interventions at the organizational level.**

| Intervention content | Examples | Study population | | | |
|---|---|---|---|---|---|
| | | 1[a] | 2[b] | 3[c] | 4[d] |
| **General health care information** | Health and hygiene information, help system; education of families and communities; decreasing stigma | Aiello et al. (2011); Klomp et al. (2020); Zhang et al. (2020) | Decosimo et al. (2017); Decosimo et al. (2019) | Morelli et al. (2019) | Weissbecker et al. (2018) |
| **Fostering sense of control/support** | Command courses; supervision | Aiello et al. (2011); Kahn et al. (2016); Klomp et al. (2011); Klomp et al. (2020); Schreiber et al. (2019); Zhang et al. (2020) | | | |
| **Preparation courses** | Debriefings; Training-of-Trainer; Preparing for Work Overseas (PFWO); Public Health Readiness Certificate Program (PHRCP) | Horn et al. (2019); Klomp et al. (2020); Waterman et al. (2019); Zhang et al. (2020) | | | |
| **Supportive leadership** | Visibility of leadership; supportive and aware leadership; valuing the contributions of frontline staff; engage staff for commitment | Aiello et al. (2011); Klomp et al. (2011) | | | |
| **Staff safety measures** | Fatigue alertness; disaster sites; protective equipment; antiviral protection; safe spaces for staff during break times | Aiello et al. (2011); Klomp et al. (2011); Klomp et al. (2020); Narra et al. (2017) | | | |
| **Providing additional family support** | Organization of child caring; involving family coping strategies; communication with families of patients; housing of family members | Aiello et al. (2011); Zhang et al. (2020) | Decosimo et al. (2017); Decosimo et al. (2019) | Morelli et al. (2019); Yoon et al. (2016) | Weissbecker et al. (2018) |
| **Dealing with death** | Dignified burials; large ceremonies; involving local religious leaders; providing services for families of deceased persons | | | | Weissbecker et al. (2018) |

[a] 1, healthcare professionals and crisis personnel

[b] 2, children and adolescents

[c] 3, community members

[d] 4, patients with highly contagious infectious disease and people at elevated risk due to somatic conditions.

studies that included a mix of providing information (e.g. pandemic-related knowledge, stress) and teaching psychological strategies (e.g. training of resilience factors and coping strategies) [21, 22, 26, 31, 32, 35, 38, 40]. Besides, relaxation and mindfulness techniques (e.g. meditation) were often included [22, 26, 31, 33–36, 40, 43, 46, 47]. Several studies [22, 39, 40, 48] indicate that psychological interventions such as CBT, or trainings including Eastern spiritual techniques like in the SMART approach [22, 40, 46], need to be adapted to the cultural context of the target group and respective concepts of mental health.

As pointed out by several studies [39, 43–45], train the trainer concepts and the provision of interventions by trained peers (e.g. other healthcare professionals) or para professionals (e.g. survivors) can be beneficial. Consistently, PFA (see Table 2) was often (partly) included in the interventions [32, 34, 35, 39, 43, 47]. However, the efficacy of PFA in general is critically discussed and was rarely examined in RCTs, especially during pandemics [32, 56]. Horn and colleagues [47] also concluded advantages of training non-specialists to provide psychosocial support during emergencies. However, participants might only benefit from short-term trainings if they already have certain skills before the training.

Highly contagious disease outbreaks and the associated changes (e.g. quarantine) as major stressors can have considerable psychosocial impacts on individuals, including short-term (e.g. stigma, anxiety) and long-term effects (e.g. trauma, grief) [e.g. 10, 57]. Since not all individuals exposed will develop mental health problems, stepped care approaches [e.g. 39] might be especially useful for mental health services during disease outbreaks.

The reviewed literature provides preliminary evidence that individuals deployed to disease outbreaks might benefit from psychological interventions, such as resilience training, offered in the pre-incident period [23, 34, 35, 37]. For example, a pre-event stress inoculation training that requires participants to create personalized resilience plans before deployment might improve the coping on-site and reduce (long-term) mental health effects of the deployment, such as posttraumatic distress. In addition to several trainings to foster the resilience of individuals in deployment situations, the Centers for Disease Control and Prevention (CDC) has implemented a pre-deployment screening process, which aims to reduce the likelihood of deploying people at risk of mental health problems [34, 35].

While the contents of nearly all included studies referred to the individual level (e.g. coping skills training, strengthening of resilience factors, relaxation training), especially the interventions targeting healthcare workers and crisis personnel also included elements at the organizational level (e.g. supportive leadership, supervision). In resilience research, a reciprocal association between individual and organizational resilience [58] has been suggested [59]. Organizational resilience (e.g. at the level of hospitals) might facilitate individual resilience in response to a stressor and vice versa. Thus, in the face of highly contagious infectious disease outbreaks, a combination of individual- and organizational-level training elements could also be important to better reach individuals and to provide them with comprehensive psychosocial and mental health support. In the context of a global pandemic like the current COVID-19 crisis, which can affect societies across different countries and regions at large, especially community interventions might benefit of organizational-level contents (e.g. fostering social support and connectedness in a community). Given the still limited number of training programs for children and adolescents, school-based interventions would be desirable (e.g. pandemic preparedness courses at class level). Finally, psychosocial and mental health support for patient populations or people at risk (e.g. due to underlying chronic diseases) might be also provided with the help of organizational structures in healthcare services, such as hospitals or general practitioners.

The findings of this scoping review, especially regarding *during-* and *after*-exposure interventions, might be of use with respect to the COVID-19 pandemic. The general public and specific risk groups, like healthcare workers and various patient populations, were mostly addressed during previous epidemic and pandemic disease outbreaks, with positive quantitative and qualitative findings reported by the respective studies. Thus, these population groups could also benefit from COVID-19 related interventions. This is in line with recent systematic reviews and meta-analyses which also found increased psychological distress and (stress-related) mental symptoms in these groups caused by the current pandemic and the associated measures of containment (e.g. quarantine) [10, 60–68]. Another practical implication of the reviewed literature refers to the intervention contents and methods used in the included studies. As discussed above, the provision of information (e.g. on infectious disease) and the strengthening of psychological coping strategies (e.g. resilience factors) were valuable approaches to provide mental health and psychosocial support in the timeline of different disease outbreaks. Given that the satisfaction with COVID-19 related knowledge [69–74] and certain resilience factors (e.g. social support) [71, 75, 76] have also been shown as protective factors during the current pandemic, these contents might serve as a first starting point or guidance for the development of health-promoting interventions in view of the COVID-19 pandemic.

Despite the positive effects of interventions that were reported in several included studies, this review reveals several evidence gaps to be addressed, with respect to specific target groups, the time point, delivery, and content of training programs (e.g. combination of individual- and organizational-level contents), and the design of intervention studies. Based on these

findings, Table 4 summarizes suggested topics for further research in the field which might also be of benefit for the COVID-19 related (intervention) research.

Regarding participants, only a minority of studies assessed interventions to cope with the potential negative impact of quarantine measures [26, 38, 42] on mental health. During disease outbreaks, people with elevated health risks are confronted with worries concerning their health as well as social stigma [22], making them an important target group of psychological interventions, which was, however, hardly represented in the literature. The same applies to older adults as well as to children and adolescents who are also exposed to various outbreak-related stressors (e.g. social isolation, homeschooling) and should be addressed–with programs adapted to their needs–in more interventions. Other system-relevant professions (e.g. grocery workers), who provide essential services for the general population, even during quarantine, were neglected in the studies. Furthermore, individuals with mental disorders deserve special attention during disease outbreaks, especially the ones who struggle to understand what is happening to them or who do not adhere to required procedures [43]. Finally, the review indicates the urgent need for more interventions to be conducted before the exposure to disease outbreaks in order to prepare communities and institutions (e.g. hospitals) for the various stressors and prevent negative mental health outcomes. Some studies particularly point to the need of trials under "real-world pandemic conditions" which would allow to measure the participants' actual response instead of proxy measures of resilience and mental health [31]. Therefore, virtual reality methods, used in two included studies [34, 35], might be a good supplement for pre-outbreak trainings. Finally, more interventions that are implemented after the exposure to disease outbreaks, such as the current pandemic, would be desirable in order to mitigate potential negative psychosocial and mental health effects.

**Table 4. Recommended research topics based on the findings of this review.**

|  | Recommended research topic |
| --- | --- |
| **Target groups of interventions** | • Individuals with elevated health risks (e.g. underlying chronic disease)<br>• Individuals with pre-existing mental disorders (e.g. depression)<br>• Employees in other critical infrastructure sectors [77] than the healthcare sector (e.g. food supply, energy, information technology and communications, transportations)<br>• Children and adolescents<br>• Older adults |
| **Time point of interventions** | More interventions |
|  | • *before* the exposure to disease outbreak (e.g. pandemic preparedness interventions)<br>• *during* the exposure:<br> • more training programs to deal with negative effects of quarantine and other measures of containment<br> • more trainings at peak of a pandemic (e.g. acute pandemic wave)<br>• *after* the exposure in order to mitigate potential long-term negative effects |
| **Delivery of interventions** | • For pre-exposure trainings: simulation trainings (e.g. using Virtual reality [VR])<br>• Low-threshold interventions using computer-based, online, mobile-based, or telephone delivery |
| **Content of interventions** | • Combination of training elements at individual and organizational level (e.g. hospitals) |
| **Study design** | • Controlled study designs (e.g. wait-list control followed by attention control group designs) and RCTs<br>• Larger sample sizes<br>• Improved study outcomes (e.g. hard outcomes such as absence from work, domestic violence, diagnoses of mental disorders based on comprehensive clinical assessment, mortality)<br>• Longer follow-up periods<br>• Dismantling designs |

Only two of the eligible completed studies focused on computer-based or online and telephone interventions. Providing low-threshold interventions that are easy to implement during acute disease outbreaks and quarantine, this delivery format should be used more frequently in future studies. The findings of current registered trials to cope with the COVID-19 pandemic will provide important evidence regarding the efficacy of this delivery format to promote mental health and psychosocial support in highly contagious emerging disease outbreaks of substantial public concern.

The overall quality of evidence found in this review was low with only two level-1 studies [22, 32]. Due to weaknesses in study designs (e.g. no control groups), some positive results may only represent natural improvements over time rather than a response to the intervention [e.g. 31]. Therefore, controlled studies and RCTs are urgently needed. As long as no standards or clearly effective interventions are available, trainings should be contrasted with non-participation to allow fair comparisons, and more hard outcomes depending on the target group (e.g. absence from work, domestic violence, development of mental disorders, mortality) and extended follow-up assessments would be desirable. Because of the mix of interventions and training methods in the included studies, the "effective component" cannot be identified [31], making dismantling designs for future studies beneficial. Nevertheless, non-RCTs and other study designs can provide key lessons of relevance for the current COVID-19 outbreak as well as future disease outbreaks (see Tables 2 and 3).

As we performed this scoping review to respond to the needs of decision makers during the current coronavirus pandemic timely, the review has limitations: the protocol was only registered at OSF in parallel, and potentially relevant trial registers and grey literature were not explored. Based on the limited evidence base, we are not able to make reliable conclusions concerning the efficacy of recommended targets in Tables 2 and 3. Limitations of the reviewed studies often include small samples sizes, few RCTs, and short follow-ups. Some interventions were not provided within the peak of a pandemic [e.g. 39, 41] but when the risk and thus psychosocial stressors had started to decrease, potentially leading to natural improvements of psychological outcomes. Conclusions made for a particular sample or disease outbreak might not apply to other target groups or disease outbreaks, and heterogeneity renders comparisons between studies difficult. Strengths of this scoping review include the hand-searching of reference lists, contacting study authors to ask for unavailable full-texts, and the inclusion of various study designs to provide the full picture of the current evidence.

## Conclusions

To our knowledge, this scoping review is the first to systematically identify and summarize the available literature on psychological interventions to foster adjustment and/or to promote mental health and psychosocial support in individuals (possibly) exposed to outbreaks of highly contagious emerging infectious diseases. Although the quality of evidence is limited due to the small number of studies and their methodological limitations, the reviewed literature reports beneficial effects for mental health and resilience factors in various target groups, with overall positive qualitative feedback from participants and intervention providers. There is an urgent need for better-designed studies, pandemic preparedness interventions, focus on specific target groups (e.g. children, specific system-relevant professions), and low-threshold and easy-access (e.g. online) interventions to address large population groups. Given the psychological burden caused by disease outbreaks and associated changes like quarantine, more research is required to inform practice and policy in order to improve mental health and psychosocial support before, during, and after outbreaks. The findings may serve as a guide for researchers and policy makers during the current COVID-19 pandemic and future highly

contagious disease outbreaks in order to plan and conduct interventions and to establish pandemic preparedness plans.

## Supporting information

**S1 Appendix. Search strategies.** Search strategies in three electronic databases.
(PDF)

**S1 Table. Eligibility criteria.** Detailed eligibility criteria of the systematic scoping review.
(PDF)

**S2 Table. PRISMA extension for Scoping Reviews (PRISMA-ScR) checklist.**
(PDF)

## Author Contributions

**Conceptualization:** Angela M. Kunzler, Klaus Lieb.

**Formal analysis:** Angela M. Kunzler, Marlene Stoll, Alexander L. Mancini, Sophie Lehmann, Manpreet Blessin, Donya Gilan, Isabella Helmreich, Klaus Lieb.

**Methodology:** Angela M. Kunzler, Jutta Stoffers-Winterling, Klaus Lieb.

**Supervision:** Klaus Lieb.

**Writing – original draft:** Angela M. Kunzler, Klaus Lieb.

**Writing – review & editing:** Angela M. Kunzler, Jutta Stoffers-Winterling, Frank Hufert, Klaus Lieb.

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
