## [Decision Letter · Decision Letter 0]

5 Oct 2020

PONE-D-20-16959

Mental health and psychosocial support strategies in highly contagious emerging disease outbreaks of substantial public concern: a systematic scoping review

PLOS ONE

Dear Dr. Lieb,

Thank you for submitting your manuscript to PLOS ONE. After careful consideration, we feel that it has merit but does not fully meet PLOS ONE’s publication criteria as it currently stands. Therefore, we invite you to submit a revised version of the manuscript that addresses the points raised during the review process.

We look forward to receiving your revised manuscript.

Kind regards,

Brita Roy, MD, MPH, MHS

Academic Editor

PLOS ONE

Journal Requirements:

2. Given that scoping reviews are mostly intended to map out a research field, and that they do not contain a detailed quality assessment, they cannot support conclusions on the effectiveness of the interventions  analysed; thus, we suggest that the conclusions reported in your Discussion section are revised accordingly.

'AMK, MS, ALM, MB and SW have no conflicts of interest. DG is in training as a board-certified cognitive-behavior (CB) therapist. JSW and IH are board-certified CB therapists. FH is board-certified in medical microbiology and virology and a specialist in tropical medicine (DTM&H) with a special interest in POCT-based rapid viral diagnostics. KL is a board-certified CB therapist with a special interest in schema therapy; he is also an Editor with the Cochrane Developmental, Psychosocial and Learning Problems Group. '

Reviewers' comments:

Reviewer's Responses to Questions

**Comments to the Author**

1. Is the manuscript technically sound, and do the data support the conclusions?

Reviewer #1: Yes

Reviewer #2: Yes

2. Has the statistical analysis been performed appropriately and rigorously? 

Reviewer #1: N/A

Reviewer #2: Yes

3. Have the authors made all data underlying the findings in their manuscript fully available?

Reviewer #1: Yes

Reviewer #2: Yes

4. Is the manuscript presented in an intelligible fashion and written in standard English?

Reviewer #1: Yes

Reviewer #2: No

5. Review Comments to the Author

Reviewer #1: This is a workmanlike (in a positive sense) scoping review of the literature, both published and grey, on strategies for coping with mental health and psychosocial issues in the face of an outbreak analogous to Covid. The methods are solid and the results are described clearly. The evident limitations, namely low volume/quality, lack of long-term follow up, and absence of digital/tech approaches are noted briefly. The paper would benefit from a table of suggested recommended research topics, now listed in prose.

Reviewer #2: This timely scoping review study seeks to summarise current mental health interventions during previous infectious disease outbreaks and and COVID-19 pandemic. The Introduction outlines the need for the study and the search strategy is comprehensive.

There are a few areas that need to be addressed by the team as follows.

1.Can the team explain why and how the rating scale for assessment of collected evidence (Oxford Centre for EBM) was modified.

2.There are several typos in the paper which should be better proof read

eg Page 13, line 127 “….concepts along with their qualitatively and quantitatively effects” etc

3.The term “diseased people’ (Page13, line 129), on page 23, in Tables 1-3 and throughout the text needs to be rephrased.

4.In Table 1, PsySTART-R needs to be included in the Abbreviations below the Table.

5.Table 1, study by Waterman et al 2018, the interventions need to be split up as it is difficult to follow.

6.Page 22, “Finally, when evaluating the same CBT group in former Ebola Treatment Center (ETC) staff, positive effects on anxiety, depression and functional impairment were improved, with the effects being independent of the exposure to risk of infection during the ETC work [40].”, the word improved should be replaced with “reported”

7.Page 27, the sentence “The quality of the evidence ranged between level 1 and level 5 (except for level-3 studies), with two level-1 studies [23, 31], two level-2 studies [30, 43, 44], 12 of level 4 [22, 32-40, 42, 45, 47] and three level-5 studies [24, 27, 41]” needs clarification and elaboration on what the levels mean and significance.

8.Page 27, the sentence “With 12 studies, healthcare professionals and crisis personnel (eg, local hospital or treatment center staff, deployed individuals providing help on-site) – ie, individuals directly exposed to acute disease outbreaks – were most frequently focused” seems awkward and needs to be rephrased.

9.The sentence on page 35 “Especially the recent activities of the Centers for Disease Control and Prevention (CDC) are worth to be mentioned here” needs to be rephrased.

10.Discussion needs to discuss the relative importance of interventions at the individual and organizational levels.

11.The team needs to include a para or two on practical applications based on extant data thus far such as methods, content and target groups which would be useful during the current pandemic.

6. PLOS authors have the option to publish the peer review history of their article (what does this mean?). If published, this will include your full peer review and any attached files.

Reviewer #1: **Yes: **David Matchar

Reviewer #2: **Yes: **Kang Sim, MD

---

## [Author Response · Author response to Decision Letter 0]

27 Nov 2020

Response to Reviewers (Please see also attached file "Response to Reviewers"

and https://journals.plos.org/plosone/s/file?id=ba62/PLOSOne_formatting_sample_title_authors_affiliations.pdf

Reply: The whole manuscript was checked for meeting the PLOS ONE style requirement, including those for file naming. Any changes made in formatting are presented in the file labeled “Revised Manuscript with Track Changes”.

2. Given that scoping reviews are mostly intended to map out a research field, and that they do not contain a detailed quality assessment, they cannot support conclusions on the effectiveness of the interventions analyzed; thus, we suggest that the conclusions reported in your Discussion section are revised accordingly.

Reply: Thank you for this comment. We have modified the Discussion section in order not to draw any conclusions about the effectiveness of the interventions analyzed. 

“Several implications for the implementation of psychological interventions in view of disease outbreaks can be derived from the reviewed literature. With this systematic scoping review, we identified interventions to provide psychosocial and mental health support in different target groups in the face of highly contagious infectious disease outbreaks. Evidence of positive effects on psychological outcomes was mostly found forreported by studies that included a mix of providing information (e.g. pandemic-related knowledge, stress) and teaching psychological strategies (e.g. training of resilience factors and coping strategies) [21, 22, 26, 31, 32, 35, 38, 40].

[…]

“The reviewed literature This reviewprovides preliminary evidence that individuals deployed to disease outbreaks domight benefit from psychological interventions, such as resilience training, offered in the pre-incident period [23, 34, 35, 37].”

[…]

“Despite the preliminary evidence forpositive effects of interventions that were reported in several included studies, this review reveals several evidence gaps to be addressed, with respect to […].”

'AMK, MS, ALM, MB and SW have no conflicts of interest. DG is in training as a board-certified cognitive-behavior (CB) therapist. JSW and IH are board-certified CB therapists. FH is board-certified in medical microbiology and virology and a specialist in tropical medicine (DTM&H) with a special interest in POCT-based rapid viral diagnostics. KL is a board-certified CB therapist with a special interest in schema therapy; he is also an Editor with the Cochrane Developmental, Psychosocial and Learning Problems Group. '

a. Please confirm that this does not alter your adherence to all PLOS ONE policies on sharing data and materials, by including the following statement: "This does not alter our adherence to PLOS ONE policies on sharing data and materials.” (as detailed online in our guide for authors http://journals.plos.org/plosone/s/competing-interests<about:blank>). If there are restrictions on sharing of data and/or materials, please state these. Please note that we cannot proceed with consideration of your article until this information has been declared.

Please know it is PLOS ONE policy for corresponding authors to declare, on behalf of all authors, all potential competing interests for the purposes of transparency. PLOS defines a competing interest as anything that interferes with, or could reasonably be perceived as interfering with, the full and objective presentation, peer review, editorial decision-making, or publication of research or nonresearch articles submitted to one of the journals. Competing interests can be financial or nonfinancial, professional, or personal. Competing interests can arise in relationship to an organization or another person. Please follow this link to our website for more details on competing interests: http://journals.plos.org/plosone/s/competing-interests

Reply: The Competing Interests statement was adapted accordingly in the cover letter by adding the statement "This does not alter our adherence to PLOS ONE policies on sharing data and materials.”

4. PLOS requires an ORCID iD for the corresponding author in Editorial Manager on papers submitted after December 6th, 2016. Please ensure that you have an ORCID iD and that it is validated in Editorial Manager. To do this, go to ‘Update my Informationʼ (in the upper left-hand corner of the main menu), and click on the Fetch/Validate link next to the ORCID field. This will take you to the ORCID site and allow you to create a new iD or authenticate a pre-existing iD in Editorial Manager. Please see the following video for instructions on linking an ORCID iD to your Editorial Manager account: https://www.youtube.com/watch?v=_xcclfuvtxQ

Reply: We have validated the ORCID iD of the corresponding author in Editorial Manager.

Reviewer #1

Reviewer #1: This is a workmanlike (in a positive sense) scoping review of the literature, both

published and grey, on strategies for coping with mental health and psychosocial issues in the face of

an outbreak analogous to Covid. The methods are solid and the results are described clearly. The

evident limitations, namely low volume/quality, lack of long-term follow up, and absence of

digital/tech approaches are noted briefly. The paper would benefit from a table of suggested

recommended research topics, now listed in prose.

Reply: Thank you for this comment. As suggested, we have added Table 4 which presents recommended research topics in a succinct and structured way in addition to the text (see also comment 11 of Reviewer #2) 

“Despite the positive effects of interventions that were reported in several included studies, this review reveals several evidence gaps to be addressed, with respect to specific target groups, the time point, delivery and content of training programs (e.g. combination of individual- and organizational-level contents), and the design of intervention studies. Based on these findings, Table 4 summarizes suggested topics for further research in the field which might also be of benefit for the COVID-19 related (intervention) research.”

Table 4. Recommended research topics based on the findings of this review.

Recommended research topic

Target groups of interventions

• Individuals with elevated health risks (e.g. underlying chronic disease)

• Individuals with pre-existing mental disorders (e.g. depression)

• Employees in other critical infrastructure sectors [77] than the healthcare sector (e.g. food supply, energy, information technology and communications, transportations)

• Children and adolescents

• Older adults

Time point of interventions

More interventions

• before the exposure to disease outbreak (e.g. pandemic preparedness interventions)

• during the exposure:

- more training programs to deal with negative effects of quarantine and other measures of containment

- more trainings at peak of a pandemic (e.g. acute pandemic wave)

• after the exposure in order to mitigate potential long-term negative effects

Delivery of interventions

• For pre-exposure trainings: simulation trainings (e.g. using Virtual reality [VR])

• Low-threshold interventions using computer-based, online, mobile-based, or telephone delivery

Content of interventions

• Combination of training elements at individual and organizational level (e.g. hospitals)

Study design

• Controlled study designs (e.g. wait-list control followed by attention control group designs) and RCTs

• Larger sample sizes

• Improved study outcomes (e.g. hard outcomes such as absence from work, domestic violence, diagnoses of mental disorders based on comprehensive clinical assessment, mortality)

• Longer follow-up periods

• Dismantling designs

\f

Reviewer #2 

Reviewer #2: This timely scoping review study seeks to summarize current mental health

interventions during previous infectious disease outbreaks and COVID-19 pandemic. The

Introduction outlines the need for the study and the search strategy is comprehensive.

There are a few areas that need to be addressed by the team as follows.

1. Can the team explain why and how the rating scale for assessment of collected evidence (Oxford Centre for EBM) was modified.

Reply: The Oxford Centre for Evidence-based Medicine for ratings of individual studies and the National Health and Medical Research Council (NHMRC) evidence hierarchy are evidence hierarchies to differ different levels of evidence from various research areas, including therapy or intervention studies. Given that the current review is a scoping review, which usually do not require a formal assessment of the methodological quality of included studies (see PRISMA extension for Scoping Reviews [PRISMA-ScR] Checklist and Peters et al., 2015 also cited in the manuscript), we still wanted to give the reader an easily understandable impression of the quality of the included studies. Therefore, we have used a modified and simple rating scheme to differ the levels of evidence among the included studies.

In the revised manuscript, we have added the following paragraph to describe our methods in more detail. In addition, in order to make the assessment easier to understand and to make each level more distinct (see also comment 7 below), we have reduced the levels assessed to four levels instead of five. We have modified Table 1 and the Results section (see comment no. 7) accordingly.

“The quality of the evidence of completed and published studies was assessed by two independent reviewers (AK, KL) using a rating scheme that was modified from the Oxford Centre for Evidence-based Medicine (EBM) for ratings of individual studies [28] and the Australian National Health and Medical Research Council (NHMRC) evidence hierarchy [29]. Given the status of this work as a scoping review, which do not require a formal assessment of the methodological quality of included studies [30], we used a rating scheme with four levels in order to assess the level of evidence (level 1: randomized controlled trial; level 2: controlled trial without randomization, case-control study or controlled time series design; level 3: single-group study with either posttest or pretest/posttest assessment; level 4: case study/report with either posttest or pretest/posttest assessment).”

2. There are several typos in the paper which should be better proof read

eg Page 13, line 127 “….concepts along with their qualitatively and quantitatively effects” etc

Reply: Thank you for your comment. The paper as a whole was checked for any typos (any changes made in file labeled “Revised Manuscript with Track Changes”.

We have corrected the above-mentioned sentence to “Instead, we aimed at giving a comprehensive overview of intervention concepts based on the available quantitative and qualitative evidence.”

3. The term “diseased peopleʼ (Page13, line 129), on page 23, in Tables 1-3 and throughout the text needs to be rephrased.

Reply: Many thanks. As suggested, we have rephrased the term throughout the manuscript: 

• Methods: “To do so, we first grouped the publications by target populations (healthcare professionals and crisis personnel; children and adolescents, community members, patients with highly contagious infectious disease, and people at elevated risk due to somatic conditions), […]”

• Table 1: “Interventions for diseased people patients with highly contagious infectious disease”

• Results: “One study focused on diseased people patients with a highly contagious infectious disease by investigating patients during the 2014 Ebola outbreak [43].”

• Table 2, Footnotes: “4, diseased people patients with highly contagious infectious disease and people at elevated risk due to somatic conditions.”

• Table 3, Footnotes: “4, diseased people patients with highly contagious infectious disease and people at elevated risk due to somatic conditions.”

• Results: “Various populations are targeted, including the general population [49, 50], diseased peopledifferent patient populations (e.g. obstetrics/gynecology patients) [51-53] or certain risk groups […]”

4. In Table 1, PsySTART-R needs to be included in the Abbreviations below the Table.

Reply: Thank you. We have added the abbreviation in the Footnotes below Table 1.

5. Table 1, study by Waterman et al 2018, the interventions need to be split up as it is difficult to

follow.

Reply: The stepped-care intervention by Waterman et al. (2018), which was also evaluated by Waterman et al. (2019) in a qualitative study with the facilitators of training (CBT component), is one intervention with three subsequent phases. As Table 1 only present different rows for studies reporting different interventions, we think it is important to present them in the same row. However, to make it easier for readers to follow, we have added bullet points in the respective row for Waterman et al. (2018) and Waterman et al. (2019).

6. Page 22, “Finally, when evaluating the same CBT group in former Ebola Treatment Center (ETC)

staff, positive effects on anxiety, depression and functional impairment were improved, with the effects being independent of the exposure to risk of infection during the ETC work [40].”, the word improved should be replaced with “reported”

Reply: Many thanks. We have replaced “improved” with “reported” in this sentence. 

7. Page 27, the sentence “The quality of the evidence ranged between level 1 and level 5 (except for level-3 studies), with two level-1 studies [23, 31], two level-2 studies [30, 43, 44], 12 of level 4 [22, 32-40, 42, 45, 47] and three level-5 studies [24, 27, 41]” needs clarification and elaboration on what the levels mean and significance.

Reply: Thank you. As suggested (see also comment 1 above), we have reduced the number of levels to four instead of five in order to make it easier to comprehend. To clarify the difference between the levels, we have included the description of the levels and have also explained them in more detail in the methods section.

Due to the changes made in the number of levels (four instead of five levels), the results presented here were modified accordingly. 

“The quality of the evidence ranged between level 1 and level 4, with two level-1 studies that were RCTs [22, 32], two (level 2) controlled studies without randomization [31, 44, 45], 11 single-group studies of level 3 [21, 33-41, 46, 48], and four case studies/reports of level 4 [23, 26, 42, 43].”

8. Page 27, the sentence “With 12 studies, healthcare professionals and crisis personnel (eg, local

hospital or treatment center staff, deployed individuals providing help on-site) – ie, individuals

directly exposed to acute disease outbreaks – were most frequently focused” seems awkward and needs to be rephrased.

Reply: Thank you for this comment. We have modified the sentence as follows: “Healthcare professionals and crisis personnel (e.g. local hospital or treatment center staff, deployed individuals providing help on-site) – that is, individuals directly exposed to acute disease outbreaks – were most frequently addressed by interventions, with 12 studies performed in this target group.”

9. The sentence on page 35 “Especially the recent activities of the Centers for Disease Control and

Prevention (CDC) are worth to be mentioned here” needs to be rephrased.

Reply: Many thanks. We have deleted the above-mentioned sentence and have included the information in the following sentence:

“Especially the recent activities of the Centers for Disease Control and Prevention (CDC) are worth to be mentioned here. In addition to several trainings to foster the resilience of individuals in deployment situations, the Centers for Disease Control and Prevention (CDC) has implemented a pre-deployment screening process, which aims to reduce the likelihood of deploying people at risk of mental health problems [34, 35].”

10. Discussion needs to discuss the relative importance of interventions at the individual and

organizational levels.

Reply: Thank you for your comment. We agree that the difference between individual- and organizational-level intervention contents and their relative importance has been neglected in the previous version of the manuscript. Therefore, we have added the following paragraph:

“While the contents of nearly all included studies referred to the individual level (e.g. coping skills training, strengthening of resilience factors, relaxation training), especially the interventions targeting healthcare workers and crisis personnel also included elements at the organizational level (e.g. supportive leadership, supervision). In resilience research, a reciprocal association between individual and organizational resilience [58] has been suggested [59]. Organizational resilience (e.g. at the level of hospitals) might facilitate individual resilience in response to a stressor and vice versa. Thus, in the face of highly contagious infectious disease outbreaks, a combination of individual- and organizational-level training elements could also be important to better reach individuals and to provide them with comprehensive psychosocial and mental health support. In the context of a global pandemic like the current COVID-19 crisis, which can affect societies across different countries and regions at large, especially community interventions might benefit of organizational-level contents (e.g. fostering social support and connectedness in a community). Given the still limited number of training programs for children and adolescents, school-based interventions would be desirable (e.g. pandemic preparedness courses at class level). Finally, psychosocial and mental health support for patient populations or people at risk (e.g. due to underlying chronic diseases) might be also provided with the help of organizational structures in healthcare services, such as hospitals or general practitioners.”

11. The team needs to include a para or two on practical applications based on extant data thus far such as methods, content and target groups which would be useful during the current pandemic.

Reply: Many thanks for this proposal. As suggested, we have added the following paragraph in order to discuss potential practical implications of the reviewed literature (e.g. target groups, intervention contents) for the COVID-19 pandemic. In line with the comment of Reviewer #2, we have also newly added Table 4.

“The findings of this scoping review, especially regarding during- and after-exposure interventions, might be of use with respect to the COVID-19 pandemic. The general public and specific risk groups, like healthcare workers and various patient populations, were mostly addressed during previous epidemic and pandemic disease outbreaks, with positive quantitative and qualitative findings reported by the respective studies. Thus, these population groups could also benefit from COVID-19 related interventions. This is in line with recent systematic reviews and meta-analyses which also found increased psychological distress and (stress-related) mental symptoms in these groups caused by the current pandemic and the associated measures of containment (e.g. quarantine) [10, 60-68]. Another practical implication of the reviewed literature refers to the intervention contents and methods used in the included studies. As discussed above, the provision of information (e.g. on infectious disease) and the strengthening of psychological coping strategies (e.g. resilience factors) were valuable approaches to provide mental health and psychosocial support in the timeline of different disease outbreaks. Given that the satisfaction with COVID-19 related knowledge [69-74] and certain resilience factors (e.g. social support) [71, 75, 76] have also been shown as protective factors during the current pandemic, these contents might serve as a first starting point or guidance for the development of health-promoting interventions in view of the COVID-19 pandemic.

Despite the positive effects of interventions that were reported in several included studies, this review reveals several evidence gaps to be addressed, with respect to specific target groups, the time point, delivery and content of training programs (e.g. combination of individual- and organizational-level contents), and the design of intervention studies. Based on these findings, Table 4 summarizes suggested topics for further research in the field which might also be of benefit for the COVID-19 related (intervention) research.”

[Table 4; see comment 1 of Reviewer #1]

---

## [Editor Report · Decision Letter 1]

16 Dec 2020

Mental health and psychosocial support strategies in highly contagious emerging disease outbreaks of substantial public concern: a systematic scoping review

PONE-D-20-16959R1

Dear Dr. Lieb,

We’re pleased to inform you that your manuscript has been judged scientifically suitable for publication and will be formally accepted for publication once it meets all outstanding technical requirements.

Kind regards,

Brita Roy, MD, MPH, MHS

Academic Editor

PLOS ONE

---

## [Editor Report · Acceptance letter]

21 Jan 2021

PONE-D-20-16959R1 

Mental health and psychosocial support strategies in highly contagious emerging disease outbreaks of substantial public concern: a systematic scoping review 

Dear Dr. Lieb:

I'm pleased to inform you that your manuscript has been deemed suitable for publication in PLOS ONE. Congratulations! Your manuscript is now with our production department. 

Kind regards, 

on behalf of

Dr. Brita Roy 

Academic Editor

PLOS ONE